# The AST/ALT (De Ritis) Ratio Predicts Survival in Patients with Oral and Oropharyngeal Cancer

**DOI:** 10.3390/diagnostics10110973

**Published:** 2020-11-19

**Authors:** Olivia Knittelfelder, Daniela Delago, Gabi Jakse, Sabine Reinisch, Richard Partl, Heidi Stranzl-Lawatsch, Wilfried Renner, Tanja Langsenlehner

**Affiliations:** 1Department of Therapeutic Radiology and Oncology, Comprehensive Cancer Center, Medical University of Graz, 8036 Graz, Austria; o.knittelfelder@hotmail.com (O.K.); daniela.delago@stud.medunigraz.at (D.D.); gabriele.jakse@klinikum-gaz.at (G.J.); richard.partl@medunigraz.at (R.P.); heidi.stranzl@medunigraz.at (H.S.-L.); tanja.langsenlehner@medunigraz.at (T.L.); 2Department of Otorhinolaryngology, Comprehensive Cancer Center, Medical University of Graz, 8036 Graz, Austria; sabine.reinisch@klinikum-graz.at; 3Clinical Institute of Medical and Chemical Laboratory Diagnostics, Medical University of Graz, 8036 Graz, Austria

**Keywords:** oral and oropharyngeal cancer, radiotherapy, aminotransferases, aspartate aminotransaminase (AST), alanine aminotransaminase (ALT), AST/ALT (DeRitis) ratio, prognostic factor, survival

## Abstract

Aminotransaminases, including aspartate aminotransaminase (AST) and alanine aminotransaminase (ALT), are strongly involved in cancer cell metabolism and have been associated with prognosis in different types of cancer. The purpose of the present study was to evaluate the prognostic significance of the pre-treatment AST/ALT ratio in a large European cohort of patients with oral and oropharyngeal squamous cell cancer (OOSCC). Data from 515 patients treated for OOSCC at a tertiary academic center from 2000–2017 were retrospectively analyzed. Levels of AST and ALT were measured prior to the start of treatment. Uni- and multivariate Cox regression analyses were applied to evaluate the prognostic value of the AST/ALT ratio for cancer-specific survival (CSS) and overall survival (OS), survival rates were calculated. Univariate analyses showed a significant association of the AST/ALT ratio with CSS (hazard ratio (HR) 1.71, 95% confidence interval (CI) 1.38–2.12; *p* < 0.001) and OS (HR 1.69, 95% CI 1.41–2.02; *p* < 0.001). In multivariate analysis, the AST/ALT ratio remained an independent prognostic factor for CSS and OS (HR 1.45, 95% CI 1.12–1.88, *p* = 0.005 and HR 1.42, 95% CI 1.14–1.77, *p* = 0.002). Applying receiver operating characteristics (ROC) curve analysis, the optimal cut-off level for the AST/ALT ratio was 1.44, respectively. In multivariate analysis, an AST/ALT ratio > 1.44 was an independent prognostic factor for poor CSS and OS (HR 1.64, 95% CI 1.10–2.43, *p* = 0.014 and HR 1.55, 95% CI 1.12–2.15; *p* = 0.008). We conclude that the AST/ALT ratio is a prognostic marker for survival in OOSCC patients and could contribute to a better risk stratification and improved oncological therapy decisions.

## 1. Introduction

Head and neck cancer is an important global public health problem, with a worldwide annual incidence of more than 450,000 new diagnoses [1,2]. Oral and oropharyngeal squamous cell carcinomas (OOSCC) comprise the majority of these cancers and represent a major cause of morbidity and mortality. OOSCC has typically been regarded as a disease of the elderly predominantly observed in men in their sixth and seventh decades after long-time abuse of tobacco and alcohol [2]. However, over the past 30 years, a shift has occurred with increasing numbers of patients worldwide being diagnosed with OOSCC at a younger age primarily as a result of chronic latent infections of the human papillomavirus (HPV) that seems to disproportionately affect younger people [3]. Oropharyngeal squamous cell carcinoma includes cancers of the tonsils, soft palate, base of the tongue, and upper lateral and posterior pharyngeal walls, whereas oral cancer comprises tumors of the lips, buccal mucosa, gingiva, front two-thirds of the tongue, floor of the mouth, hard palate and retromolar trigone.

Despite substantial treatment advances, OOSCC remains a lethal disease for over 50% of patients, primarily because patients develop loco-regional recurrence and/or metastatic disease [2]. Recently, cancer research has focused on the characterization of novel markers, which could help to identify patients at higher risk for disease recurrence and death and select those who will benefit from specific therapies [4,5].

Commonly used blood-based parameters are relatively easy to assess, making them attractive biomarkers for improved individualized risk assessment [6,7,8]. Aminotransaminases, such aspartate aminotransaminase (AST) and alanine aminotransaminase (ALT), are expressed in different cellular subcompartments by non-cancerous and cancerous cells and are strongly involved in cellular metabolism and cancer cell turnover [9]. ALT is involved in the ‘glucose-alanine cycle’ and interchanges alanine and pyruvate and strongly contributes to the regeneration of glucose consumed by muscle. AST is essential for aerobic glycolysis by relocation of nicotinamide adenine dinucleotide (NADH) within the mitochondria. These transaminase reactions are particularly important in muscle and liver cells, but also in other cells with high metabolic activity [10]. Both AST and ALT represent easily measureable, blood-based biomarkers that are routinely analyzed before initiation of treatment.

The AST/ALT ratio, also termed the De Ritis ratio, is used to differentiate varying causes of liver disease and has also been proven to be a useful biomarker in many non-hepatic diseases [11]. Cumulating evidence suggests that the AST/ALT ratio might represent an independent prognostic factor in several cancer entities [12,13,14,15,16].

Recently, an elevated AST/ALT ratio has also been associated with poor survival in a population of patients with locally advanced head and neck squamous cell carcinoma (HNSCC) [17]. However, data on the role of the AST/ALT ratio in patients with OOSCC are limited. The aim of the present study was to evaluate the association of the pre-treatment AST/ALT ratio with cancer-specific survival (CSS) and overall survival (OS) in OOSCC patients in order to further clarify the prognostic significance of the pre-treatment plasma AST/ALT ratio.

## 2. Materials and Methods 

The study population comprised 515 patients with primary OOSCC who were treated with radiotherapy at our hospital between January 2000 and October 2017. Oral and oropharyngeal cancer patients who met the following inclusion criteria were eligible for the present study: (a) histologically confirmed primary squamous cell carcinoma; (b) no evidence of distant metastasis; (c) radiotherapy for curative intent; (d) no evidence of other malignancies; (e) no history of previous radiotherapy for HNSCC, and (f) availability of pretreatment AST and ALT levels. Patients were excluded from the present investigation if they did not meet all of the criteria mentioned above. An abnormal liver function was not considered as an exclusion criterion. From the 677 patients who had been treated for OOSCC with radiotherapy between January 2000 and October 2017, 5 patients were excluded because of evidence of distant metastases at the time of diagnosis, 13 patients because of secondary malignancies, 144 patients were excluded because pretreatment AST and ALT levels were not available. Finally, 515 patients were included in the present study.

A total of 269 patients were treated with postoperative RT, among them, 125 received concomitant chemotherapy. The remaining patients were treated with definitive radiochemotherapy or radiotherapy alone. Definitive radiotherapy was combined with docetaxel, cisplatin, 5-fluorouracil (TPF)-based induction chemotherapy in 70 patients. A total of 192 patients received concurrent chemotherapy, mostly consisting of a platinum-based regimen, although targeted therapy such as cetuximab was also used.

Pre-therapeutic plasma AST and plasma ALT were routinely analyzed by standard clinical testing methodology in lithium heparin plasma (AST and ALT upper reference level 35 and 45 U/L, respectively). Measurement of plasma AST and ALT was performed on a Roche Modular analyzer from January 2000 to December 2007, and on a Roche Cobas 8000 analyzer from January 2008 to October 2017, respectively. Accuracy of the tests was controlled using internal reference samples according to the manufacturer’s instructions as well as regular participation in external quality control schemes. In 65 patients, HPV status was determined using p16 expression detected by immunohistochemistry, additionally, HPV DNA polymerase chain reaction (PCR) was performed in 6 patients. Clinical staging was performed according to the 7th edition of the American Joint Committee on Cancer (AJCC) staging in oral and oropharyngeal cancer.

### 2.1. Ethical Considerations

The study complied with the Declaration of Helsinki and was performed according to the national law. The study protocol has been approved by the local Ethical Committee (approval number: EK 29-273 ex 16/17 from 31 March 2017 and EK 31-061 ex 18/19 from 04 December 2018). As this is a retrospective non-interventional study, the institutional review board waived the need for written informed consent from the participants.

### 2.2. Radiation Technique

All patients received radiotherapy with 6 MV photon linear accelerators. The dose-fractionation regimen was either standard fractionation or a simultaneous integrated boost (SIB) protocol. Standard fractionation radiotherapy was delivered up to a total dose of 70.0 Gy in 35 fractions (2.0 Gy per fraction/5x per week). The prescription dose to primary lesions or positive nodes ranged from 66 to 70 Gy, prophylactic nodal areas were irradiated at doses of 50 Gy. The SIB radiation schedules consisted of 5 × 2 Gy or 5 × 2.2 Gy/week to 70 Gy or 70.4 Gy to clinically manifest sites of gross disease and 5 × 1.6 Gy or 1.69 Gy/week to 56 Gy or 54 Gy to adjacent lymphatic drainage regions at risk for subclinical metastasis. 

The radiation method was either three-dimensional conformal radiotherapy or intensity modulated radiotherapy (IMRT), including volumetric modulated arc therapy (VMAT). Patients treated with postoperative radiotherapy received standard fractionation RT up to a total dose of 60–70 Gy at 2 Gy per fraction, depending on risk factors such as resection margin and tumor stage.

### 2.3. Staging und Follow-up

Clinical follow-up after the end of radiotherapy was performed both at the Department of Therapeutic Radiology and Oncology and at the Department of Otorhinolaryngology according to institutional guidelines. Physical examination was performed every 3 months (first two years), every 6 months (years 3–5), and annually thereafter. Imaging was performed as indicated by clinical examination. Loss to follow-up rate was 32% after 5 years, primarily, because the patients decided not to continue follow-up examinations at our institution and to receive medical support solely by their primary healthcare institutions and general practitioners.

### 2.4. Statistical Analysis

The primary endpoint was CSS defined as the time from the first day of treatment to the date of OOSCC cancer-related death. Overall survival (OS), the secondary endpoint, was calculated from first day of treatment to death of any cause. The relationship between pre-treatment AST/ALT levels and other clinico-pathological features was studied by non-parametric tests. Receiver operating characteristic (ROC) curve analysis was performed to estimate the optimal AST/ALT cutoff value to differentiate patient survival and cancer-related death, as reported previously [18,19,20]. The optimal cutoff value was determined as the point on the ROC curve that maximizes the Youden Index. Cox proportional hazards analysis was performed to calculate the hazard ratio (HR) and 95% confidence interval (CI) to evaluate the influence of the pre-treatment AST/ALT ratio and other clinical parameters on clinical endpoints. Multivariate Cox proportion analysis was performed to determine the influence of potential confounders and included variables significantly associated with CSS and OS in univariate analysis. Patients’ clinical end points were calculated using the Kaplan–Meier method and compared by the log-rank test.

All statistical analyses were performed using the Statistical Package for Social Sciences version 25.00 (SPSS Inc., Chicago, IL, USA) and MedCalc, version 19.0 (http://www.medcalc.com/, accessed from 14 April 2019 to 23 April 2019). A two-sided *p* < 0.05 was considered statistically significant.

## 3. Results

A total of 515 OOSCC patients were included in the present analysis. The median age at time of diagnosis was 58 years (mean 59.2 ± 10.9 years). Patient characteristics are displayed in Table 1.

The mean pre-treatment values of AST and ALT levels were 29.2 ± 17.7 U/L and 28.6 ± 22.8 U/L, respectively. The median pre-treatment AST/ALT ratio was 1.14 (mean 1.26 ± 0.66). 

The pre-treatment AST/ALT ratio significantly correlated with smoking status (*p* = 0.016), alcohol consumption (<0.001), tumor stage (*p* = 0.010), and nodal involvement (*p* = 0.047). No significant associations were found between the AST/ALT ratio and age, sex, and tumor grade (all *p* > 0.05).

In patients treated with primary radio (chemo-) therapy, the median pre-treatment AST/ALT ratio was 1.26 (mean 1.40 ± 0.68), in patients treated with primary surgery, the median pre-treatment AST/ALT ratio was 1.06 (mean 1.16 ± 0.58, *p* < 0.001, Mann–Whitney U-test) suggesting that patients with a higher AST/ALT ratio were more commonly treated with primary radio(chemo-) therapy. 

In patients treated with primary surgery, the median pre-surgical, post-surgical, and post-radiotherapy AST/ALT ratio was 1.06 (mean 1.16 ± 0.58), 1.01 (mean 1.13 ± 0.69), and 1.16 (mean 1.24 ± 0.49), respectively. In patients treated with induction chemotherapy, the median AST/ALT ratio prior to chemotherapy, after chemotherapy, and after completion of radio(chemo-) therapy was 1.26 (mean 1.34 ± 0.54), 1.60 (mean 1.67 ± 0.70) and 1.26 (mean 1.41 ± 0.59), respectively. In patients treated with definitive radio(chemo-) therapy (without induction chemotherapy), the median pre- and post- treatment AST/ALT ratio was 1.26 (mean 1.41 ± 0.72) and 1.18 (mean 1.31 ± 0.59).

Median follow-up time was 61 months (95% CI 53.8 to 68.2 months). During this period, 130 patients (25.2%) died due to OOSCC, a total of 196 patients (38.1 %) died of any cause. 

In univariate analysis, the pre-treatment AST/ALT ratio was significantly associated with CSS (HR 1.71, 95% CI 1.38–2.12; *p* < 0.001). Furthermore, univariate analysis identified tumor site, body mass index (BMI), smoking status, alcohol consumption, tumor stage, surgical resection, and induction chemotherapy as significant prognostic factors for CSS. After adjustment for tumor site, BMI, smoking status, alcohol consumption, tumor stage, surgical resection, and induction chemotherapy, the association between an elevated pre-treatment AST/ALT ratio and CSS remained statistically significant in multivariate analysis (HR 1.45, 95% CI 1.12–1.88, *p* = 0.005, Table 2). 

In OS analysis, the pre-treatment AST/ALT ratio significantly correlated with survival in univariate analysis (HR 1.69, 95% CI 1.41–2.02; *p* < 0.001, Table 3) which also showed a significant relationship between smoking status, alcohol consumption, tumor stage, surgical resection and OS. After inclusion of BMI, smoking status, alcohol consumption, tumor stage, and surgical resection in multivariate analysis, the pre-treatment AST/ALT ratio remained significantly associated with OS (HR 1.42, 95% CI 1.14–1.77, *p* = 0.002; Table 3). 

Furthermore, the prognostic value of the AST/ALT ratio measured at other time points (after surgery, after induction chemotherapy, and after radio (chemo-) therapy) was analyzed. We detected that the post-radiotherapy AST/ALT ratio was significantly associated with CSS (HR 1.57, 95% CI 1.11–2.21; *p* = 0.011) and OS (HR 1.38, 95% CI 1.04–1.84; *p* = 0.027). In addition, the AST/ALT ratio measured after induction chemotherapy was significantly associated with OS (HR 1.46, 95% CI 1.004–2.13; *p* = 0.048) whereas a significant association with CSS was not observed (HR 1.39, 95% CI 0.90–2.16; *p* = 0.140). A significant relationship between the post-surgical AST/ALT ratio and CSS (HR 1.19, 95% CI 0.86–1.65; *p* = 0.302) or OS (HR 1.19, 95% CI 0.91–1.54; *p* = 0.201) was not detected.

In the subgroup of patients with oropharyngeal cancer, the pre-treatment AST/ALT ratio was identified as significant predictor of CSS (HR 1.94, 95% CI 1.42–2.66; *p* < 0.001) and OS (HR 1.88, 95% CI 1.48–2.39; *p* < 0.001). In oral cancer patients, the pre-treatment AST/ALT ratio was also significantly associated with CSS (HR 1.52, 95% CI 1.09–2.11; *p* = 0.013) and OS (HR 1.51, 95% CI 1.12–2.04; *p* = 0.007).

Abnormal liver function was diagnosed in 60 patients (11.6%). Among these patients, the pre-treatment AST/ALT ratio was not identified as a prognostic factor for CSS (HR 0.55, 95% CI 0.17–1.79; *p* = 0.318) and OS (HR 0.77, 95% CI 0.36–1.65; *p* = 0.503).

Additionally, in the subgroup of patients treated with primary radio(chemo-)therapy, the pre-treatment AST/ALT ratio was significantly associated with CSS (HR 1.64, 95% CI 1.23–2.18; *p* = 0.001). After adjustment for parameters significantly associated with CSS in univariate analysis, the pre-treatment AST/ALT ratio remained a significant predictor of CSS in multivariate analysis (HR 1.45, 95% CI 1.05–2.00; *p* = 0.023). In patients undergoing primary radio(chemo-) therapy, a significant association between the pre-treatment AST/ALT ratio and OS has also been detected in univariate analysis (HR 1.61, 95% CI 1.260–2.07; *p* < 0.001) as well as multivariate analysis (HR 1.42, 95% CI 1.07–1.88; *p* = 0.015).

Using ROC curve analysis, a pre-treatment AST/ALT cutoff value of 1.44 was determined for both CSS (sensitivity = 38.5%; specificity = 76.9%; corresponding AUC = 0.579; Figure 1A) and OS (sensitivity = 37.1%; specificity = 79.1%; corresponding AUC = 0.592; Figure 1B).

Pre-treatment AST/ALT values equal to or below the obtained cutoff point were considered low (n = 373 patients (72.4%)), whereas AST/ALT values above the cutoff point were defined as high (n = 142 patients (27.6%)). Kaplan–Meier curves of patient CSS and OSS revealed that an increased pre-treatment AST/ALT ratio was associated with poor prognosis in patients with OOSCC (*p* < 0.001, Figure 2 and Figure 3). 

Furthermore, univariate analysis identified the pre-treatment AST/ALT ratio > 1.44 as a prognostic factor for poor CSS (HR 2.24, 95% CI 1.57–3.20) and OS (HR 2.06, 95% CI 1.53–2.75; *p* < 0.001). To determine the independent prognostic value of the pre-treatment AST/ALT cutoff for CSS and OS, we performed multivariate analysis using a Cox proportional hazard model including factors significantly associated with prognosis in univariate analysis. In multivariate analysis, the AST/ALT ratio > 1.44 remained an independent prognostic factor for both decreased CSS and OS (HR 1.64, 95% CI 1.10–2.43, *p* = 0.014 and HR 1.55, 95% CI 1.12–2.15; *p* = 0.008).

In a subgroup of patients (n = 65), information on HPV status was available (Table 1). HPV status was determined in 55 patients with oropharyngeal cancer and in 10 patients with oral cancer. In HPV negative patients, the median pre-treatment AST/ALT ratio was 1.21 (mean 1.33 ± 0.53), in HPV positive patients, the median AST/ALT ratio was 1.06 (mean 1.09 ± 0.43; *p* = 0.053) suggesting a trend for higher pre-treatment AST/ALT ratios in patients with HPV negative tumors.

The analysis of the relationship between the pre-treatment AST/ALT ratio and prognosis showed a significant association of the AST/ALT ratio > 1.44 with decreased CSS (HR 4.52, 95% CI 1.10–18.53; *p* = 0.036) and OS (HR 3.72, 95% CI 1.11–12.44, *p* = 0.033) among patients with HPV negative tumors. In patients with HPV positive tumors, a significant association between the AST/ALT ratio and CSS and OS was not detected. 

## 4. Discussion

Despite recent progress in the identification of genetic, epigenetic and common molecular alterations, lack of standardization as well as the expensive and/or time-consuming nature of the assays limit their routine use in daily clinical practice. Regularly used blood-based parameters are relatively easy to assess without additional laborious efforts, making them attractive parameters for an improved individualized risk assessment. In the present study, we evaluated the prognostic significance of the pre-treatment AST/ALT ratio in patients with OOSCC and detected a significant association between an elevated AST/ALT ratio and poor CSS and OS.

To the best of our knowledge, we are the first to describe these results in a European cohort of patients with OOSCC. The major strength of our study is the large cohort of 515 patients. To our knowledge, this study provides the largest study population investigating the association between the AST/ALT ratio and prognosis. Another strength of our study is the relatively long follow up period.

Blood-based measurement of the pre-treatment AST/ALT ratio was described as a useful prognosticator for hepatocellular carcinoma several years ago. More recently, a relationship between the AST/ALT ratio and prognosis has been shown in various tumor entities such as non-metastatic renal cancer, bladder cancer, upper tract urothelial cancer, and prostate cancer [12,13,14,15,16,20].

Bezan et al. reported that the preoperative AST/ALT ratio was an independent prognostic factor for metastasis-free survival and overall survival in patients with non-metastatic renal cancer [20]. In addition, the results from Tan et al. showed that the AST/ALT ratio was an independent prognosticator of poor survival in patients with distal cholangiocarcinoma [21]. Lindmark et al. also found that AST and ALT were significantly associated with patient survival after analyzing 212 patients with colorectal cancer [22]. Stocken et al. analyzed the data of 653 patients with advanced pancreatic cancer and concluded that AST was independently related to CSS [23]. Moreover, Nishikawa et al. detected that the AST/ALT ratio was a significant prognostic biomarker for extravesical recurrence-free survival patients with upper tract urothelial carcinoma [12].

In OOSCC, data on the prognostic role of the AST/ALT ratio are limited. To the best of our knowledge, to date only one study has examined the prognostic role of the AST/ALT ratio for head and neck cancer. Takenaka and colleagues evaluated the association of the AST/ALT ratio with overall survival in a cohort of HNSCC patients that included 156 OOSCC patients in the training set and 102 OOSCC patients in a validation set [17]. Similar to our findings, an elevated AST/ALT ratio independently predicted of increased mortality, in addition, the ratio was also used to subdivide patients with Union for International Cancer Control (UICC) stage IVA into low- and high-risk groups. However, the study cohort comprised a rather heterogeneous patient population and also included patients with laryngeal and hypopharyngeal cancers. In previous studies, AST and ALT values have also been reported to be higher in HNSCC patients when compared with healthy individuals [24]. Furthermore, it has been shown that AST and ALT values decreased and approached normal levels in cancer patients with no disease activity. The association of the pre-treatment AST/ALT ratio with tumor stage in the present investigation may support these findings.

Recently, tumor metabolism has received increasing attention in regard to the carcinogenesis of several malignancies. Warburg was among the first to demonstrate accelerated cancer cell metabolism involving enhanced aerobic glycolysis and pyruvate production. Recent studies also demonstrated increased glutamine metabolism to maintain nucleotide biosynthesis and synthesis of non-essential amino acids in proliferating cancer cells. ALT, which catalyzes the conversion of pyruvate and glutamate to alanine and α-ketoglutarate, functions in glycolysis as well as in the metabolism of glutamine [9]. In vitro experiments revealed decreased ALT levels in more invasive cells compared with less-invasive cancer cells suggesting that a decrease in serum ALT level could be a manifestation of enhanced metabolism and an increased consumption of the ALT in aggressive cancer cells [25].

AST also has a major role in aerobic glycolysis and is widely expressed in different tissue types whereas the ALT is considered more liver-specific [10]. Pathological processes associated with higher proliferation, tissue damage and tumor cell turnover may, therefore, lead to a stronger increase in AST when compared to ALT making the AST/ALT ratio an attractive potential biomarker.

The present study shows that an elevated pre-treatment AST/ALT ratio might be a prognostic factor for poor cancer-specific and overall survival in OOSCC patients. In subgroup analysis, we detected a significant association between the pre-treatment AST/ALT ratio and prognosis in patients with HPV negative tumors but not in patients with HPV positive tumors. However, the explanation for this finding remains speculative. HPV positive cancer represents a distinct clinical and biologic entity with many unresolved issues that will be investigated in future research.

Some limitations of our study should be taken into account. Because of its retrospective design, we are unable to exclude the possibility of an unequal distribution of unidentified clinico-pathologic parameters in our patient cohort that may have biased the observed results. Furthermore, information on the HPV status was available in only 65 patients.

Nevertheless, even considering these limitations, our data support the hypothesis that the pre-treatment AST/ALT ratio might represent an independent prognostic factor in OOSCC patients.

However, validation of our data in additional prospective studies is imperative to draw firm conclusions about the role of AST/ALT ratio for OOSCC prognosis. If confirmed by additional studies, determination of the AST/ALT ratio might contribute to the identification of patients who might be candidates for additional, more aggressive treatment approaches or more stringent follow-up schedules. 

In conclusion, an increased pre-treatment AST/ALT ratio seems to significantly impact prognosis in non-metastatic OOSCC patients and may support oncological therapy decisions. However, further large-scale prospective multi-centric studies are warranted to confirm and extend these findings.

## Figures and Tables

**Figure 1 diagnostics-10-00973-f001:**
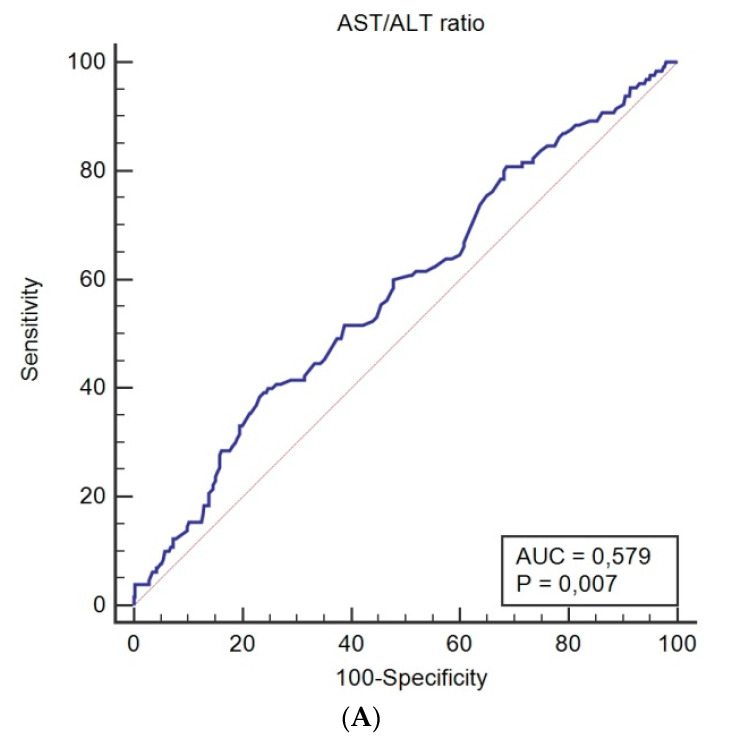
Receiver operating characteristic curves for pre-treatment AST/ALT ratio predicting cancer-specific survival (**A**), and overall survival (**B**).

**Figure 2 diagnostics-10-00973-f002:**
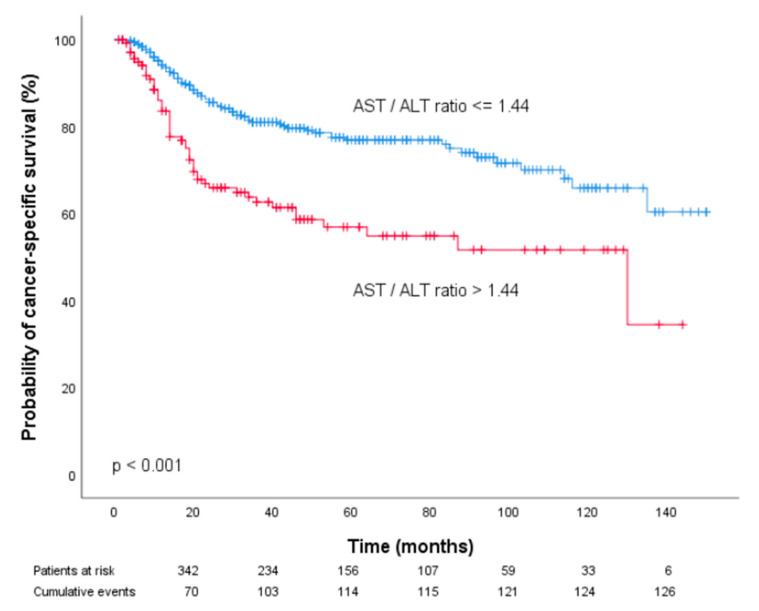
Kaplan–Meier curves for cancer-specific survival categorized by the pre-treatment AST/ALT ratio.

**Figure 3 diagnostics-10-00973-f003:**
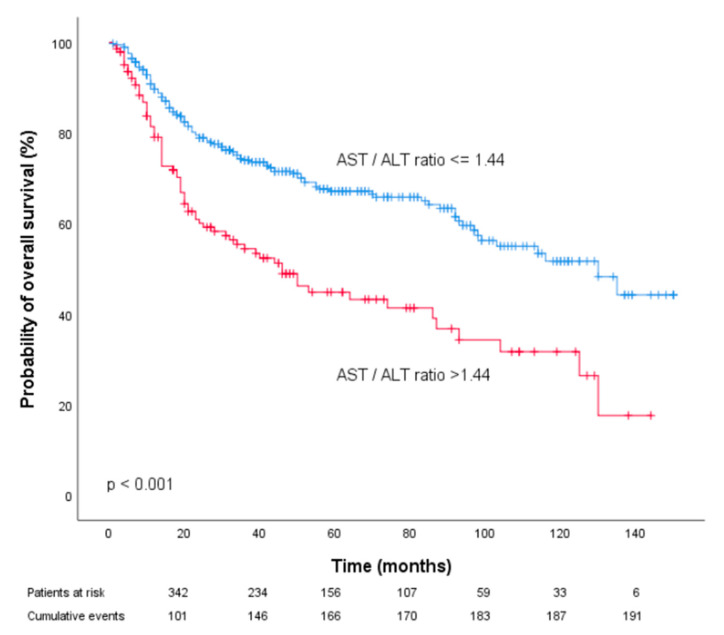
Kaplan–Meier curves for overall survival of prostate cancer patients categorized by the pre-treatment AST/ALT ratio.

**Table 1 diagnostics-10-00973-t001:** Baseline patient characteristics. Abbreviations: n = number of patients; SD = standard deviation; AST = aspartate aminotransaminase; ALT = alanine aminotransaminase; HPV = human papilloma virus; UICC = Union for International Cancer Control.

	Patient Characteristics, n (%)
Sex	
Male	388 (75.3%)
Female	127 (24.7%)
Age at diagnosis	
<60	279 (54.2)
>60	236 (45.8%)
Body mass index	
Median (mean ± SD)	24.1 (24.6 ± 4.42)
Smoking status	
Former or never *	188 (36.5%)
Current **	320 (62.1%)
Missing data	7 (1.4%)
Alcohol consumption	
Former or never *	309 (60.0%)
Current **	191 (37.1%)
Missing data	15 (2.9%)
Primary site	
Oral cavity	206 (40%)
Oropharynx	309 (60%)
Tumor grade	
G 1/2	252 (48.9%)
G 3/4	256 (49.7%)
Missing data	7 (1.4%)
Tumor stage	
T 1/2	215 (41.7%)
T 3/4	290 (56.3%)
Missing data	10 (1.9%)
HPV status	
Negative	30 (5.8%)
Positive	35 (6.8%)
Missing data	450 (87.4%)
Nodal involvement	
N0	100 (19.4%)
N+	409 (79.4%)
Missing data	6 (1.2%)
UICC stage	
I	9 (1.7%)
II	30 (5.8%)
III	99 (19.1%)
IV	372 (71.8%)
Missing data	8 (1.5%)
Primary treatment	
Surgery	269 (52.2%)
Radio(chemo-) therapy	246 (47.8%)
Induction Chemotherapy	
Yes	76 (14.8%)
No	439 (85.2%)
Concomitant Chemotherapy	
Yes	197 (38.3%)
No	317 (61.6%)
Pre-treatment AST/ALT ratio	
Median (mean ± SD)	1.14 (1.26 ± 0.66)

* Former smoking or alcohol consumption was defined as tobacco or alcohol abuse before or until the start of treatment. ** Current smoking or alcohol consumption was defined as continuation of tobacco or alcohol abuse after start of treatment.

**Table 2 diagnostics-10-00973-t002:** Univariate and multivariate analysis of clinical-pathological parameters for the prediction of cancer-specific survival. Abbreviations: CI = confidence interval; HR = hazard ratio; AST = aspartate aminotransaminase; ALT = alanine aminotransaminase; UICC = Union for International Cancer Control.

	Cancer Specific Survival			
	Univariate Analysis		Multivariate Analysis *	
	HR (95% CI)	*p*-Value	HR (95% CI)	*p*-Value
Sex				
Male	1			
Female	1.22 (0.83–1.79)	0.307		
Age at diagnosis				
<60	1			
>60	1.15 (0.81–1.63)	0.446		
Body mass index (continuous)	0.93 (0.89–0.98)	0.004	0.98 (0.93–1.03)	0.358
Smoking status				
Former/never	1		1	
Current	1.48 (1.00–2.17)	0.048	1.17 (0.75–1.83)	0.493
Alcohol consumption				
Former/never	1		1	
Current	1.76 (1.24–2.51)	0.002	1.47 (0.99–2.18)	0.058
Primary site				
Oropharynx	1		1	
Oral cavity	1.69 (1.19–2.40)	0.002	1.93 (1.32–2.82)	0.001
Tumor grade				
G 1/2	1			
G 3/4	0.99 (0.70–1.42)	0.984		
Tumor stage				
T 1/2	1		1	
T 3/4	2.20 (1.50–3.23)	<0.001	1.24 (0.77–2.01)	0.379
Nodal involvement				
N0	1			
N+	1.16 (0.73–1.86)	0.532		
UICC stage				
I	1			
II	0.74 (0.14–3.81)	0.716		
III	0.82 (0.19–3.54)	0.794		
IV	1.28 (0.31–5.18)	0.733		
Primary treatment				
Radio(chemo-) therapy	1		1	
Surgery	0.36 (0.25–0.52)	<0.001	0.34 (0.21–0.55)	<0.001
Induction chemotherapy				
No	1		1	
Yes	1.72 (1.13–2.63)	0.012	1.05 (0.73–1.51)	0.798
Concomitant chemotherapy				
No	1			
Yes	1.01 (0.70–1.44)	0.974		
Pre-treatment				
AST/ALR ratio (continuous)	1.71 (1.38–2.12)	<0.001	1.45 (1.12–1.88)	0.005

* Adjustment for all factors significantly associated with cancer-specific survival in univariate analysis.

**Table 3 diagnostics-10-00973-t003:** Univariate and multivariate analysis of clinical-pathological parameters for the prediction of overall survival. Abbreviations: CI = confidence interval; HR = hazard ratio; AST = aspartate aminotransaminase; ALT = alanine aminotransaminase; UICC = Union for International Cancer Control.

	Overall Survival			
	Univariate Analysis		Multivariate Analysis *	
	HR (95% CI)	*p*-Value	HR (95% CI)	*p*-Value
Sex				
Male	1			
Female	0.97 (0.70–1.35)	0.866		
Age at diagnosis				
<60	1			
>60	1.17 (0.88–1.56)	0.284		
Body mass index (continuous)	0.93 (0.90–0.97)	<0.001	0.96 (0.92–1.00)	0.048
Smoking status				
Former/never	1		1	
Current	1.55 (1.13–2.13)	0.006	1.19 (0.82–1.72)	0.359
Alcohol consumption				
Former or never	1		1	
Current	1.83 (1.37–2.45)	<0.001	1.52 (1.10–2.12)	0.011
Primary site				
Oropharynx	1			
Oral cavity	1.16 (0.87–1.55)	0.308		
Tumor grade				
G 1/2	1			
G 3/4	0.98 (0.73–1.30)	0.862		
Tumor stage				
T 1/2	1		1	
T 3/4	1.91 (1.41–2.59)	<0.001	1.04 (0.71–1.53)	0.845
Nodal involvement				
N0	1			
N+	1.13 (0.77–1.65)	0.53		
UICC stage				
I	1			
II	0.65 (0.17–2.51)	0.529		
III	0.91 (0.28–2.96)	0.87		
IV	1.20 (0.38–3.76)	0.759		
Primary treatment				
Radio(chemo-) therapy	1		1	
Surgery	0.42 (0.31–0.56)	<0.001	0.44 (0.30–0.64)	<0.001
Induction chemotherapy				
No	1			
Yes	1.40 (0.97–2.01)	0.073		
Concomitant chemotherapy				
No	1			
Yes	1.05 (0.78–1.41)	0.746		
Pre-treatment				
AST/ALR ratio (continuous)	1.69 (1.41–2.02)	<0.001	1.42 (1.14–1.77)	0.002

* Adjustment for all factors significantly associated with overall survival in univariate analysis.

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
