# Peer review of "The AST/ALT (De Ritis) Ratio Predicts Survival in Patients with Oral and Oropharyngeal Cancer"

_diagnostics, 2020, doi:10.3390/diagnostics10110973_

Round 1

Reviewer 1 Report

In this article, the authors evidenced the association between AST/ALT ratio and the survival of oral and oropharyngeal cancer. The rationale and the study design are acceptable and there are some comments:

  1. The introduction should be condensed and some issues can be moved to the discussion section.
  2. The critical point is the timing of AST/ALT data obtained. Is the AST/ALT data of post-operation, post-chemo, and post-RT related to the clinical outcome?
  3. Please provide the changes of AST/ALT ratio from pretreatment, post-op, post-chemo, and post-RT.
  4. Since patients with abnormal liver function was not excluded, how many patients had abnormal liver function? Is AST/ALT ration an independent prognostic marker in patients with abnormal liver function?
  5. Why the groping of tumor grade and T stage were cut in median instead of 1 vs 2+3+4. Does that make different results?
  6. Why not use the total stage which combines TNM?
  7. Oropharynx and oral cavity cancer might be different. Does the result remain the same in each of them?

Reviewer 2 Report

Dear authors,

you present a study about the prognostic impact of the AST/ALT ratio in oral and oropharyngeal cancer.

I am pretty interested in this study as we found similiar results in our head and neck cancer patients (from all localisations). So I fully support publication of your article.

However I would suggest some major revisions before publication.

1) regarding the p16 positive patients:

-> are all 65 patients where p16 status is available oropharyngeal cancer patients? In oral cancer p16 is not that relevant. So this issue should be explained in more detail. Furthermore did you compare the AST/ALT ratio between p16 positive and negative patients in general? This results should also be mentioned as p16 positive oropharyngeal cancer patients usually do not have the typical noxa anamnesis .

-> Did you also perform HPV DNA PCR analysis in the p16 positive patients? If yes please mention.

-> I would also include these patient data in table 1 (number of p16 positive/negative/ missing)

2) Did you observe any difference regarding OS and CSS after adjustment for first-line treatment in the multivariate analysis? Are patients with higher AST/ALT ratio more commenly treated with first-line radiochemotherapy/radiotherapy? Is the AST/ALT ratio still prognostic relevant if only first-line radio-/chemotherapy patients are analysed?

Round 2

Reviewer 1 Report

The authors revised the MS accordingly.

Reviewer 2 Report

Dear authors,

as you have answered all my comments thoroughly, I suggest acceptance of the article in the present form.

Good luck for the publication process.